# Nuclear-Magnetic-Resonance-Spectroscopy-Derived Serum Biomarkers of Metabolic Vulnerability Are Associated with Disability and Neurodegeneration in Multiple Sclerosis

**DOI:** 10.3390/nu16172866

**Published:** 2024-08-27

**Authors:** Taylor R. Wicks, Irina Shalaurova, Richard W. Browne, Anna Wolska, Bianca Weinstock-Guttman, Robert Zivadinov, Alan T. Remaley, James D. Otvos, Murali Ramanathan

**Affiliations:** 1Department of Pharmaceutical Sciences, State University of New York, Buffalo, NY 14214, USA; trwicks@buffalo.edu; 2LabCorp Diagnostics, Morrisville, NC 27560, USA; 3Biotechnical and Clinical Laboratory Sciences, State University of New York, Buffalo, NY 14214, USA; 4Lipoprotein Metabolism Laboratory, National Heart, Lung and Blood Institute, National Institutes of Health, Bethesda, MD 20892, USA; 5Department of Neurology, State University of New York, Buffalo, NY 14203, USA

**Keywords:** metabolic vulnerability, lipoproteins, nuclear magnetic resonance, cholesterol, branched-chain amino acids

## Abstract

Purpose: Metabolic vulnerabilities can exacerbate inflammatory injury and inhibit repair in multiple sclerosis (MS). The purpose was to evaluate whether blood biomarkers of inflammatory and metabolic vulnerability are associated with MS disability and neurodegeneration. Methods: Proton nuclear magnetic resonance spectra were obtained from serum samples from 153 healthy controls, 187 relapsing–remitting, and 91 progressive MS patients. The spectra were analyzed to obtain concentrations of lipoprotein sub-classes, glycated acute-phase proteins, and small-molecule metabolites, including leucine, valine, isoleucine, alanine, and citrate. Composite indices for inflammatory vulnerability, metabolic malnutrition, and metabolic vulnerability were computed. MS disability was measured on the Expanded Disability Status Scale. MRI measures of lesions and whole-brain and tissue-specific volumes were acquired. Results: Valine, leucine, isoleucine, alanine, the Inflammatory Vulnerability Index, the Metabolic Malnutrition Index, and the Metabolic Vulnerability Index differed between healthy control and MS groups in regression analyses adjusted for age, sex, and body mass index. The Expanded Disability Status Scale was associated with small HDL particle levels, inflammatory vulnerability, and metabolic vulnerability. Timed ambulation was associated with inflammatory vulnerability and metabolic vulnerability. Greater metabolic vulnerability and inflammatory vulnerability were associated with lower gray matter, deep gray matter volumes, and greater lateral ventricle volume. Conclusions: Serum-biomarker-derived indices of inflammatory and metabolic vulnerability are associated with disability and neurodegeneration in MS.

## 1. Introduction

Metabolic dysfunction is a central pathophysiological mechanism in multiple sclerosis (MS) [1]. Proper functioning of metabolic pathways is required for restoring blood–brain barrier integrity, resolving inflammatory lesions, and repairing damaged myelin [1]. Metabolism is also critically important for the homeostasis of immune responses, neuronal functions, and regulatory, repair, and regeneration mechanisms in the central nervous system (CNS).

Inflammatory conditions strongly modulate metabolism in immune cells and in tissues primarily via cytokines, such as tumor necrosis factor-a (TNF-a), interleukin-1β (IL-1), and interleukin-6 (IL-6), and mediators, such as nitric oxide (NO). Pro-inflammatory conditions promote a preference for glycolysis, whereas the presence of anti-inflammatory cytokines, such as IL-4, drives peripheral immune cells to increase oxidative phosphorylation and fatty acid oxidation. Inflammatory cells, such as Th1 and Th17 lymphocytes, M1 macrophages, and activated dendritic, endothelial, and microglial cells have high rates of glycolysis, and regulatory cells, such as regulatory T cells and M2 macrophages, utilize fatty acid oxidation [2,3]. TNF-a, IL-1, and IL-6 also mediate chronic inflammation, the production of acute phase proteins, and reduced albumin levels.

Carbohydrate, lipid, and amino acid metabolism are inter-connected. Citrate, a tricarboxylic acid that is a central metabolite in the Krebs cycle, also promotes the formation of prostaglandins, which are potent inflammatory lipids. Likewise, disruption of the Krebs cycle in macrophages causes the accumulation of succinate and the induction of IL-1, IL-6, and NO [4]. During immunological stress, amino acids are redistributed towards tissues involved in inflammation and the immune response to support proliferation and cytokine production [5]. Leucine, isoleucine, and valine, which are categorized as branched-chain amino acids (BCAAs), increase the translocation of glucose transporters to the cell surface in muscle cells and provide Coenzyme-A intermediates to the Krebs cycle [6]. BCAAs are associated with dyslipidemia and reduced high-density lipoprotein (HDL) cholesterol [7]. The intersections and orchestration between metabolic pathways and inflammation may be crucial in MS because blood–brain barrier breakdown and inflammatory lesions are pathological hallmarks of MS [8].

Nuclear magnetic resonance (NMR) lipoprotein profiling was initially established as a clinical research tool for cardiovascular and metabolic disease pathophysiology because it can measure the size profiles and particle concentration of lipoproteins in different sub-classes of VLDL, LDL, and HDL [9] that cannot be obtained using biochemical methods. Altered size distributions in LDL and HDL sub-compartments are associated with cardiovascular risk, diabetes, hypertriglyceridemia, and obesity [10,11,12,13,14,15]. Lipids have been implicated in MRI and disability in MS [16]. NMR spectroscopic profiling has now been further advanced to include accurate measurements of BCAAs, citrate, and glycated inflammatory proteins (GlycAs). However, the interdependence among BCAAs, citrate, and GlycAs have not been investigated in MS. NMR spectroscopy is a powerful approach for investigating the interactions between different arms of nutrient metabolism and its intersections with chronic inflammation. Validation of a blood-based NMR–spectroscopic measure may allow for individualized, biochemically targeted clinical interventions that could potentially help address metabolic vulnerability in MS.

Three indices for inflammation, metabolic malnutrition, and metabolic vulnerability were developed by Otvos et al. from the NMR profile [17]. The Inflammatory Vulnerability Index (IVX) is obtained from GlycA and small HDL particles (sHDLP). The Metabolic Malnutrition Index (MMX) includes BCAA and citrate. The Metabolic Vulnerability Index (MVX) combines the IVX and MMX scores and accounts for the synergistic interaction in malnutrition–inflammation syndromes [17]. The goal of this research was to investigate the association of the IVX, MMX, and MVX with clinical factors, disability, and MRI measures in MS.

## 2. Methods

### 2.1. Study Design

Study Design and Setting: An observational, case-controlled, prospective study of clinical, genetic, and environmental risk factors in MS provided the clinical and biomarker data for this study. The study was conducted at the Jacobs Multiple Sclerosis Center at the University at Buffalo, an academic health science center in Buffalo, NY.

Informed Consent: The study protocol was approved by The University at Buffalo Human Subjects Institutional Review Board. Written informed consent was provided by all subjects.

Clinical Assessments: Demographics (e.g., age, sex, etc.) were obtained at baseline. Subjects also provided blood samples and underwent neurological examinations.

MS diagnosis was based on the 2010 revision of the McDonald criteria [18]. MS disease course was categorized as relapsing–remitting (RR), secondary progressive (SP), and primary progressive (PMS) using accepted disease course classifications [19]. The exclusion criteria included clinically defined relapse, use of corticosteroids in the preceding 30 days, and pregnant or nursing mothers.

Expanded Disability Status Scale (EDSS) scores were used to measure disability [20]. The EDSS is an ordinal measure of MS disability that ranges from 0 to 10 in 0.5 increments. An EDSS score of 0 corresponds to no disability, EDSS 3.0 is moderate disability, individuals with an EDSS of 6.0 require unilateral assistance (e.g., a cane) to walk, individuals with EDSS 8.0 are restricted to a bed, chair, or wheelchair, and EDSS 10 represents death due to MS. Timed ambulation was obtained from a timed 25-foot walk (T25FW). Information regarding the history of heart problems was obtained from participants’ self-report to a disease history questionnaire.

Healthy controls (HCs) needed to meet health screening requirements and have a normal physical and neurological examination. HCs were recruited with frequency matching in mind.

Non-fasting blood samples were processed to separate serum within 24 h. The serum samples were frozen in aliquots at −80 °C.

This sub-study was restricted to HC subjects and MS patients who were ≥18 years old with NMR-derived biomarker panels available. The progressive MS (PMS) group comprised SPMS and PPMS.

MRI Image Acquisition: Patients underwent brain MRI on a 3-T General Electric Signa 4x/Lx, scanner (General Electric, Boston, MA, USA). Axial dual fast spin-echo (FSE) T2/PD-weighted image (WI), 3D-spoiled-gradient recalled (SPGR) T1-WI, spin echo (SE) T1-WI with and without gadolinium (Gd) contrast, and fluid-attenuated inversion recovery (FLAIR) scans were acquired.

MRI Image Analysis: The MRI analysts were blinded to patients’ clinical characteristics and clinical status.

T2- and T1-hypointense lesion volumes (LVs) were obtained using a semi-automated edge-detection contouring–thresholding technique previously described [21].

The SIENAX cross-sectional software tool was used, with correction for T1-hypointensity misclassification, for brain extraction and tissue segmentation [22]. Normalized measures of whole brain volume (WBV), gray matter volume (GMV), deep gray matter volume (DGMV), cortical volume (CV), and lateral ventricle volume (LVV) were computed as previously described [23].

### 2.2. Serum Nuclear Magnetic Resonance (NMR) Analysis

Proton NMR spectra (LipoScience, Morrisville, NC, USA) [22] were obtained from serum samples. The proton NMR spectra were analyzed with the LP4 (or LipoProfile-4) deconvolution algorithm [17,24,25] to obtain concentrations of different-sized sub-classes of triglyceride-rich lipoproteins, LDL, and HDL particles, mean particle sizes of these lipoprotein classes, and several small-molecule metabolites, including leucine, valine, isoleucine, alanine, and citrate. In addition, GlycA, an NMR-derived biomarker of inflammation that corresponds to a signal from glycan residues of acute-phase glycoproteins, was measured. The concentration of metabolites is expressed in µM.

The sHDLP sub-class was defined as HDL particles with a size < 9 nm [17].

The branched-chain amino acid (BCAA) concentration was defined as the sum of valine, leucine, and isoleucine concentrations.

IVX, MMX, and MVX were computed using the following formulae:
*IVX* = 9 − 0.0027 *GlycA* − 0.46079 *sHDLP* + 0.0006325 *GlycA* × *sHDLP**IVXmin* = 2.0 ⟹ *score* = 1*IVXmax* = 8.3 ⟹ *score* = 100*MMX* = 0.75097(4 − 0.02234 *Leu* + 0.0000528 *Leu*^2^) + 0.55737(7 − 0.02895 *Val* +  0.0000608 *Val*^2^) + (0.00867 *Ile*) + 0.65649(1 + 0.0025 *Cit* + 0.0000167 *Cit*^2^)*MMXmin* = 1.281 ⟹ *score* = 1*MMXmax* = 2.0 ⟹ *score* = 100*MVX* = 2.72923 *IVX* + 11.96062ln(*MMX*) − 1.12749 *IVX* × ln(*MMX*) *MVXmin* = 20.3 ⟹ *score* = 1*MVXmax* = 28.0 ⟹ *score* = 100

*Leu*, *Val*, *Ile*, and *Cit* refer to leucine, valine, isoleucine, and citrate levels, respectively. The subscripts *min* and *max* refer to the minimum and maximum values of the corresponding index. The quadratic fits for *Leu*, *Val*, *Ile*, and *Cit* and the natural logarithm transformation (*ln*) of (*MMX*) improved the fit. The coefficients were obtained from Cox proportional hazards fit to the 3-year mortality in the CATHGEN data [19].

### 2.3. Data Analysis

The R statistical program was used [26] on a MacBook Pro computer running macOS Catalina (10.15.7). The results were visualized using the *ggplot2* R package (Version 3.5.1) [27].

HC-RRMS-PMS status was a categorical variable denoting the corresponding groups. Body mass index (BMI) was computed as the ratio of weight (in kg) to the square of height (in meters). Obesity was defined as BMI ≥ 30 kg/m^2^. Heart problem status was a binary variable denoting the presence or absence of heart problems. Tertiles of IVX, MMX, and MVX were computed in the RR-MS and PMS groups using the ntile function in R.

The differences in baseline demographic, clinical, and biomarker variables between the HC and RR-MS groups and the PMS groups were assessed using the *t*-test for continuous variables, the chi-square test for categorical variables, or the Fisher exact test for binary variables.

The NMR-derived biomarkers (i.e., leucine, valine, isoleucine, alanine, citrate, sHDLP, BCAA, IVX, MVX, and MMX) were individually assessed as dependent variables in regression analyses with HC-RRMS-PMS status, age, sex, and BMI as predictors. To assess the effects of heart problems, additional regression analyses of the NMR-derived biomarkers (as dependent variables) were conducted with HC-RRMS-PMS status, age, sex, BMI, and heart problem status as predictors.

Timed ambulation was logarithm (base 10)-transformed for regression analyses. NMR-derived biomarkers were individually assessed as dependent variables in regression analyses with age, sex, BMI, and either EDSS or timed ambulation.

Also, the NMR-derived biomarkers were individually assessed as dependent variables in regression analyses with age, sex, BMI, and individual MRI measures (i.e., T2-LV, T1-LV, WBV, GMV, DGMV, CV, or LVV).

The *rstatix* R package was used to compute the generalized eta-squared ($\eta^{2}$, a measure of effect size) [28] and the predictor *p*-value for regression analyses [29]. The $\eta^{2}$ thresholds for small, medium, or large effect sizes are ≥0.01, 0.06, and 0.14, respectively [30].

## 3. Results

Clinical and Demographic Characteristics: The clinical and demographic characteristics of the study groups are summarized in Table 1. The PMS group was older and had greater EDSS than the RR-MS group. These differences are representative of the respective disease courses.

NMR Biomarkers in MS: The regression results for the HC-RR-PMS predictor are summarized in Table 2 for each serum NMR-derived biomarker. The regression analyses were adjusted for age, sex, and BMI.

Valine, leucine, isoleucine, BCAA, and alanine were negatively associated with HC-RR-PMS status (Table 2), i.e., the levels were lower in the RR and PMS groups vs. HC. GlycA, IVX, MMX, and MVX were positively associated with HC-RR-PMS status, i.e., the levels were higher in the RR and PMS groups vs. HC.

Figure 1 graphically shows the patterns of dependence with HC-RR-PMS status. The mean valine, leucine, isoleucine, alanine, and BCAA levels were lower in the RR-MS and PMS groups compared to HC. GlycA exhibited an increasing pattern, with PMS > RR-MS > HC. The mean values of IVX were greater in RR-MS and PMS relative to HC, whereas MMX and MVX had the increasing pattern of PMS > RR-MS > HC.

The frequencies of heart problems in the HC, RR, and PMS groups were 30 (20%), 43 (23%), and 20 (22%), respectively. Because some NMR biomarkers are associated with cardiovascular diseases, we conducted regression analyses with age, sex, BMI, HC-RR-PMS status, and heart problem status variables. The associations of valine, leucine, isoleucine, alanine, BCAA, GlycA, IVX, MMX, and MVX with HC-RR-PMS status remained significant upon adjusting for the presence of heart problems (Appendix A).

Associations with Disability Measures: The regression results for EDSS and timed ambulation predictors in MS are summarized in Table 3 for each NMR-derived biomarker. The regression analysis also adjusted for age, sex, and BMI. Disability on the EDSS was negatively associated with sHDLP and positively associated with IVX and MVX. Ambulation time was negatively associated with alanine and positively associated with IVX and MVX.

Figure 2 graphically summarizes the dependence of EDSS and ambulation time on the tertiles of IVX, MMX, and MVX. The mean values of EDSS increased with increasing tertiles of IVX and MMX. Ambulation time was greater at the higher quartiles of IVX. The mean values of EDSS and ambulation time were greater at the highest quartiles of MVX vs. the lower quartiles.

Associations with MRI Measures: The regression results for the MRI measures, T2-LV, T1-LV, WBV, GMV, DGMV, CV, and LVV with individual NMR biomarkers are summarized in Table 4. The regression analysis was adjusted for age, sex, and BMI. Valine, leucine, and BCAA were associated with both T2-LV and T1-LV. Valine was also associated with GMV, CV, and LVV. IVX and MVX were associated with GMV, DGMV, and LVV. MVX was additionally associated with T2-LV.

Figure 3 graphically summarizes the dependence of T2-LV, GMV, DGMV, and LVV on the tertiles of MVX in the RR-MS and PMS groups. T2-LV was greater at the higher tertiles of MVX relative to the lowest tertile in PMS (Figure 3A); no dependence was found for RR-MS. GMV and DGMV were lower in the higher tertiles of MVX relative to the lowest tertile in RR-MS (Figure 3B,C). In PMS, GMV was lower in the highest tertile of MVX, whereas DGMV was lower with increasing tertiles in PMS. LVV was greater with increasing tertiles of MVX for PMS; no differences were noted for RR-MS (Figure 3D).

## 4. Discussion

We investigated the associations of blood-based biomarkers linked to inflammatory and metabolic vulnerability with disability and neurodegeneration in MS. We found that levels of BCAA and the composite indices, IVX, MMX, and MVX, were associated with HC-RRMS-PMS status. Disability on the EDSS was associated with IVX and MVX and a low level of sHDLP. Metabolic systems for lipids, amino acids, and carbohydrates are inter-connected and co-regulated to maintain energy homeostasis; as such, the IVX, MMX, and MVX indices are useful as composite biomarkers of these metabolic pathways.

NMR spectroscopic profiling methods can simultaneously measure levels of multiple lipoproteins, amino acids, and carbohydrate metabolites in serum samples. NMR-based approaches measure concentrations non-destructively and are not prone to the differential extraction and ionization efficiency issues that affect the accuracy of mass spectrometric liquid chromatography detectors that are frequently used in metabolomic studies [31]. NMR profiling was also selected because it has been widely investigated in clinical studies [19,32,33]. The investigation of the IVX, MMX, and MVX for MS was motivated by the known codependence of metabolomics and inflammation in the general population and in MS [34,35]. The independent and combined contributions of IVX, MMX, and MVX were associated with all-cause mortality in patients who underwent heart catheterization [19]. There are sex differences in IVX, MMX, and MVX resulting from sex differences in metabolite levels, but these are unrelated to mortality risk [19]. Upon conducting regression analyses that adjusted for sex, age, and BMI, we found that IVX and MVX had associations with disability and multiple MRI brain atrophy measures in MS.

Overall, IVX is an emerging biomarker already linked to CVD mortality risk that might concomitantly represent systemic inflammation in MS. The GlycA component of IVX is a biomarker of inflammation that is linked to cardiometabolic risk [32], and it represents a composite of NMR signals from the glycosylation of multiple acute-phase proteins [33,36]. GlycA was increased in RR-MS, and PMS compared to HC. We found that sHDLPs, which also contribute to IVX, were negatively associated with EDSS. However, we could not confirm the increases in sHDLP in RR-MS vs. HC reported by Jorissen et al. [37].

We also found lower levels of BCAA, leucine, isoleucine, and valine in MS compared to HC. Additionally, there were association trends (*p* < 0.15) for greater EDSS and ambulation time with lower BCAAs in MS. BCAAs are essential amino acids, and, unlike other amino acids that undergo hepatic metabolism, BCAAs are metabolized in muscle and promote muscle anabolism [38]. Decreases in serum valine, leucine, and isoleucine in MS [39] and alterations to BCAA pathways that are enriched in metabolomic studies have been reported by other groups [40,41]. The decreases in BCAAs can cause diminished muscle mass and functions and may contribute to MS disability.

A strength of our study was the availability of MRI biomarkers of MS lesions (e.g., T2-LV, T1-LV), including global (WBV), tissue-specific (GMV), and central atrophy (LVV) biomarkers that are more sensitive than clinical measures, such as the EDSS. MVX was associated with T2 lesions, which are a measure of MS disease burden. The pathology of T2 lesions can range from reversible demyelination and edema to irreversible damage, e.g., axonal loss. MMX was associated with T1-LV, which assesses hypointensities from lesions with persistent tissue destruction. Atrophy is the end result of neurodegeneration in MS. Gray matter (GM) lesions in the cerebral cortex [42,43] contribute to cortical atrophy and decreased CV, whereas lesions in deep gray matter (DGM) structures, such as the cerebellum, basal ganglia, and thalamus, contribute to DGM atrophy [34]. MVX and IVX were associated with GMV and DGMV and exhibited a trend with CV. MVX and IVX were also associated with increased LVV, which represents central atrophy or ventricular volume expansion following brain tissue loss [35].

In addition to whole-brain and tissue-specific atrophy, MS pathology can also cause changes in ‘normal-appearing’ white matter. A limitation is that we did not assess the association of our NMR measures with other MRI modalities, e.g., diffusion tensor imaging (DTI), magnetization transfer ratio (MTR), and functional MRI (fMRI), which can provide insights into MS pathophysiological mechanisms [44]. DTI measures, such as fractional anisotropy, radial axial, and mean diffusivity, can be useful because intact axons and their myelin sheets constrain the flow of water to a preferred direction of flow. Decreased fractional anisotropy and changes in radial and axial diffusivity indicate axonal loss and demyelination, respectively. It can detect decreased microstructural integrity of both lesional and normal-appearing white matter. Magnetization transfer imaging can also detect pathological changes in normal-appearing white matter. The magnetization transfer ratio is a neuroimaging measure of the integrity of myelin and other cell membranes. Lower MTR indicates greater white matter pathology, such as astrocytic hyperplasia and axonal damage [45].

In conclusion, we found changes in IVX and MVX, which are composite indices of inflammatory and metabolic vulnerability in MS vs. HC. IVX and MVX were also associated with greater disability and neurodegeneration in MS. Future studies should conduct further validation of the NMR-derived composite indices and their clinical associations. Validated NMR-spectroscopy-derived composite biomarkers of inflammatory and metabolic vulnerability might be useful for assessing disability, nutritional factors, inflammatory processes, and frailty in MS.

## Figures and Tables

**Figure 1 nutrients-16-02866-f001:**
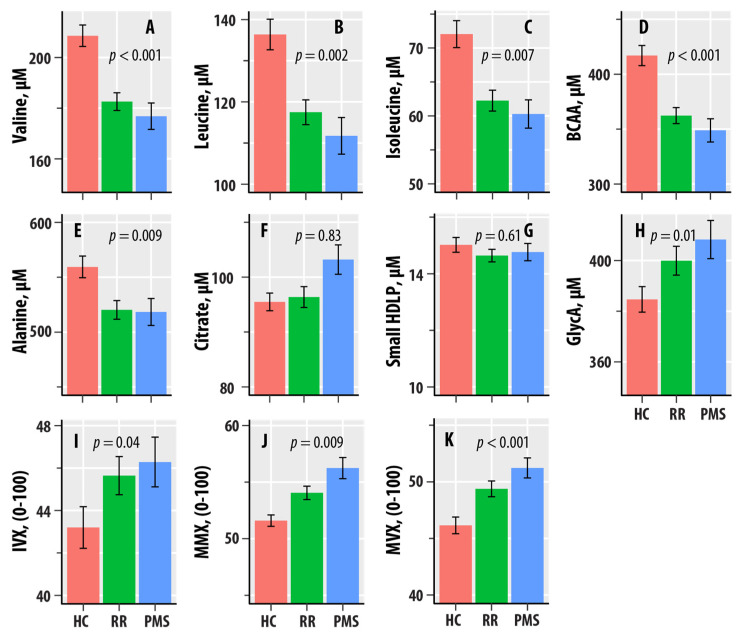
(**A**–**I**) are bar graphs summarizing results from nuclear magnetic resonance spectroscopic profiling in healthy controls (HCs, salmon bars), relapsing–remitting multiple sclerosis (RR, green bars), and progressive MS (PMS, blue bars). The levels of valine (**A**), leucine (**B**), isoleucine (**C**), total branched-chain amino acids (BCAAs, **D**), alanine (**E**), citrate (**F**), small HDL particles (**G**), and glycated proteins (GlycA, **H**) are in µM. The values of the Inflammation Vulnerability Index (IVX, **I**), the Metabolic Malnutrition Index (MMX, **J**), and the Metabolic Vulnerability Index (MVX, **K**) range from 0 to 100. The bars represent mean values, and the error bars are standard errors. The regression *p*-values from Table 1 are shown.

**Figure 2 nutrients-16-02866-f002:**
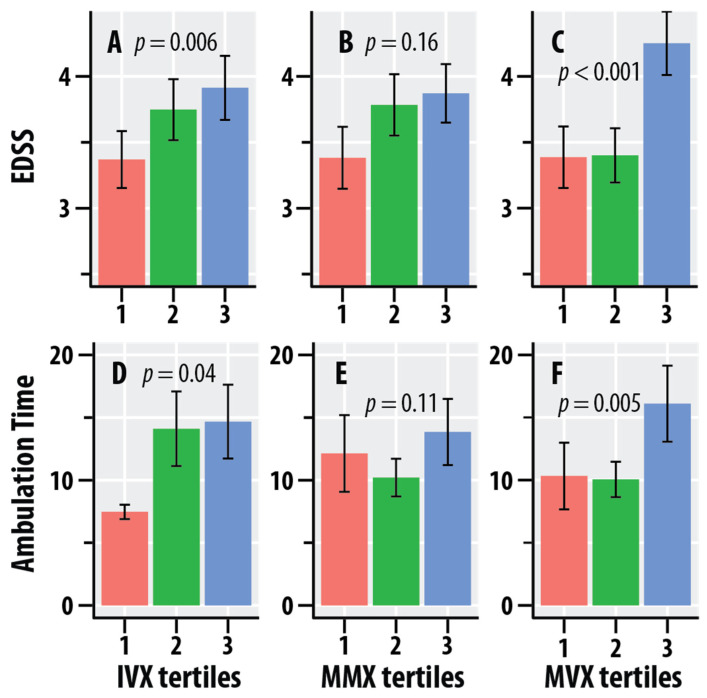
Dependence of the Expanded Disability Status Scale (EDSS, **A**–**C**) scores and timed 25-foot walk scores (ambulation time, seconds, **D**–**F**) on tertiles of the Inflammation Vulnerability Index (IVX, **A**,**D**), Metabolic Malnutrition Index (MMX, **B**,**E**), and Metabolic Vulnerability Index (MVX, **C**,**F**). The salmon bars represent the lowest tertile, the green bars represent the intermediate tertile, and the blue bars are the highest tertile; the tertiles were calculated in the MS group. The bars represent mean values, and the error bars are standard errors. The regression *p*-values for IVX, MMX, and MVX from Table 3 are shown.

**Figure 3 nutrients-16-02866-f003:**
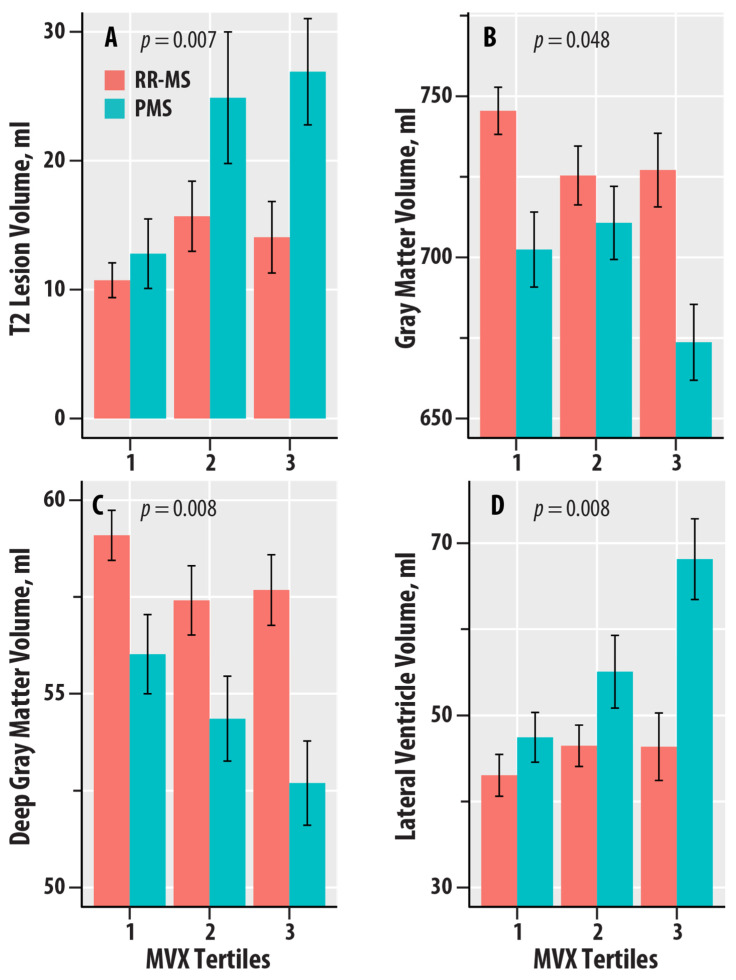
Dependence of T2 lesion volume (in ml, **A**), gray matter volume (in ml, **B**), deep gray matter volume (in ml, **C**), and lateral ventricle volume (in ml, **D**) on tertiles of the Metabolic Vulnerability Index (MVX) in relapsing–remitting MS (RR-MS, salmon bars) and progressive MS (PMS, teal bars). The bars represent mean values, and the error bars are standard errors. The regression *p*-values for IVX, MMX, and MVX from Table 4 are shown.

**Table 1 nutrients-16-02866-t001:** Demographic and clinical characteristics by disease course at baseline.

	HC	RR-MS	P-MS	*p*-Value
Sample size *n*	153	187	91	–
Gender, female (%)	85 (55.6)	141 (75.4)	66 (72.5)	<0.001
Age, years	45.9 (13.9)	44.3 (9.65)	54.1 (8.82)	<0.001
Body mass index, kg/m^2^	28.0 ± 5.94	27.1 ± 6.15	25.6 ± 5.44	<0.001
Race:				
Caucasian	133 (88.1%)	171 (92.4%)	86 (94.5%)	–
African American	13 (8.6%)	9 (4.9%)	4 (4.4%)	
Hispanic/Latino	1 (.66%)	3 (1.6%)	1 (1.1%)	
Asian	3 (2.0%)	1 (.54%)	–	
Other	1 (0.66%)	1 (.54%)	–	
Missing	2 (1.3%)	2 (1.1%)	–	
Disease duration, years	–	12.1 (8.47)	21.5 (11.4)	<0.001
EDSS ^a^	-	2.5 (1.5-3.5)	6.0 (5-6.5)	<0.001
Disease-modifying treatments:				
No treatment	–	20 (12.0%)	16 (18.2%)	–
Interferon		67 (40.4%)	31 (35.2%)	
Glatiramer acetate		35 (21.1%)	25 (28.4%)	
Other		44 (26.5%)	16 (18.2%)	
Missing		21	3	

^a^ All of the continuous variables are mean ± standard deviation. EDSS is summarized as median (inter-quartile range). *p*-values for the differences between the HC and RR-MS groups were based on the independent samples *t*-test. The frequency of females to males was based on the Fisher exact test. EDSS: Expanded Disability Status Scale. Interferon-beta category included AVONEX^®^ and REBIF^®^. The Other category included natalizumab, mitoxantrone, intravenous immunoglobulin, and mycophenolate mofetil.

**Table 2 nutrients-16-02866-t002:** Results from the regression analysis of NMR-derived biomarkers with HC-RR-PMS status. All regression analyses were adjusted for age, sex, and BMI. The generalized eta-squared ($\eta^{2}$) effect size measure (*p*-value) is shown; $\beta_{RR}$ and $\beta_{PMS}$ are regression coefficients for the RR and PMS groups relative to HC.

NMR Biomarker	HC-RR-PMS $\eta^{2}$ (*p*-Value)	$\beta_{RR}$	$\beta_{PMS}$
Valine	0.036 (<0.001)	−18.2	−21.6
Leucine	0.031 (0.002)	−14.5	−19.5
Isoleucine	0.025 (0.007)	−6.92	−8.01
BCAA	0.038 (<0.001)	−39.6	−49.1
Alanine	0.023 (0.009)	−35.9	−44.1
Citrate	<0.001 (0.83)	0.965	2.03
sHDLP	0.002 (0.61)	−0.176	−0.439
GlycA	0.023 (0.01)	16.4	27.5
IVX	0.016 (0.04)	1.85	4.22
MMX	0.023 (0.009)	1.87	3.21
MVX	0.039 (<0.001)	2.42	4.96

Abbreviations: BCAAs: branched-chain amino acids; sHDLP: small high-density lipoprotein particles; GlycA: glycated proteins; IVX: Inflammatory Vulnerability Index; MMX: Metabolic Malnutrition Index; MVX Metabolic Vulnerability Index.

**Table 3 nutrients-16-02866-t003:** Results from the regression analysis of NMR-derived biomarkers with EDSS or timed ambulation as predictors in MS. All regression analyses adjusted for age, sex, and BMI. The generalized eta-squared ($\eta^{2}$) effect size measure (*p*-values) and regression coefficient β are shown.

NMR Biomarker	EDSS		Timed Ambulation *	
	$\eta^{2}$ (*p*-Value)	β	$\eta^{2}$ (*p*-Value)	β
Valine	0.010 (0.11)	−2.36	0.012 (0.12)	−16.4
Leucine	0.010 (0.13)	−2.04	0.016 (0.08)	−16.5
Isoleucine	0.014 (0.06)	−1.23	0.016 (0.08)	−8.28
BCAA	0.013 (0.07)	−5.62	0.017 (0.07)	−41.1
Alanine	0.010 (0.12)	−5.72	0.031 (0.015)	−63.0
Citrate	0.002 (0.52)	0.534	<0.001 (0.69)	2.29
sHDLP	0.022 (0.02)	−0.225	0.007 (0.25)	−0.764
GlycA	0.009 (0.14)	3.43	0.010 (0.16)	23.2
IVX	0.031 (0.006)	1.02	0.023 (0.04)	5.13
MMX	0.008 (0.16)	0.379	0.013 (0.11)	2.97
MVX	0.046 (<0.001)	0.946	0.041 (0.005)	5.34

* Timed ambulation was logarithm-transformed. Abbreviations: BCAAs: branched-chain amino acids; sHDLPs: small high-density lipoprotein particles; GlycAs: glycated proteins; IVX: Inflammatory Vulnerability Index; MMX: Metabolic Malnutrition Index; MVX Metabolic Vulnerability Index.

**Table 4 nutrients-16-02866-t004:** Results from the regression analysis of NMR-derived biomarkers with age, sex, BMI, and different MRI measures as predictors. The generalized eta-squared value ($\eta^{2}$) effect size measure and *p*-values for the MRI measures are shown.

NMR-Derived Biomarker	T2-LV $\eta^{2}$ (*p*-Value)	T1-LV $\eta^{2}$ (*p*-Value)	WBV $\eta^{2}$ (*p*-Value)	GMV $\eta^{2}$ (*p*-Value)	DGM $\eta^{2}$ (*p*-Value)	CV $\eta^{2}$ (*p*-Value)	LVV $\eta^{2}$ (*p*-Value)
Valine	0.020 (0.04)	0.020 (0.04)	0.012 (0.10)	0.020 (0.03)	0.015 (0.07)	0.024 (0.02)	0.017 (0.046)
Leucine	0.024 (0.02)	0.035 (0.006)	0.001 (0.60)	0.001 (0.58)	0.005 (0.30)	0.003 (0.41)	0.011 (0.11)
Isoleucine	0.012 (0.10)	0.019 (0.046)	0.014 (0.07)	0.015 (0.06)	0.014 (0.07)	0.019 (0.04)	0.013 (0.09)
BCAA	0.024 (0.02)	0.031 (0.01)	0.008 (0.17)	0.011 (0.10)	0.012 (0.09)	0.016 (0.06)	0.017 (0.046)
Alanine	<0.001 (0.83)	0.005 (0.32)	0.003 (0.41)	0.001 (0.59)	0.004 (0.34)	0.002 (0.50)	<0.001 (0.96)
Citrate	0.001 (0.62)	0.003 (0.46)	0.008 (0.18)	0.001 (0.63)	0.007 (0.19)	0.002 (0.49)	<0.001 (0.94)
sHDLP	0.015 (0.06)	0.009 (0.16)	0.001 (0.63)	0.005 (0.28)	0.017 (0.047)	0.003 (0.37)	0.005 (0.27)
GlycA	0.004 (0.35)	<0.001 (0.94)	0.008 (0.18)	0.015 (0.065)	0.013 (0.08)	0.013 (0.08)	0.023 (0.022)
IVX	0.015 (0.07)	0.002 (0.55)	0.005 (0.27)	0.017 (0.048)	0.028 (0.01)	0.014 (0.08)	0.019 (0.04)
MMX	0.014 (0.08)	0.019 (0.04)	<0.001 (0.93)	<0.001 (0.88)	<0.001 (0.69)	<0.001 (0.79)	0.006 (0.25)
MVX	0.032 (0.007)	0.014 (0.08)	0.004 (0.32)	0.017 (0.048)	0.030 (0.008)	0.015 (0.06)	0.030 (0.008)

Abbreviations: BCAAs: branched-chain amino acids; sHDLPs: small high-density lipoprotein particles; GlycAs: glycated proteins; IVX: Inflammatory Vulnerability Index; MMX: Metabolic Malnutrition Index; MVX Metabolic Vulnerability Index. T2-LV: T2-weighted lesion volume, ml; T1-LV: T1-weighted lesion volume, ml; WBV: whole brain volume; GMV: gray matter volume; DGM: deep gray matter volume; CV: cortical volume; LVV: lateral ventricle volume, mL.

## Data Availability

The data that support the findings of this study are available upon reasonable request from the principal investigator of the clinical study (Dr. Robert Zivadinov). The data is not publicly available due to containing information that could compromise the privacy of research participants.

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
