# Peer review of "Nuclear-Magnetic-Resonance-Spectroscopy-Derived Serum Biomarkers of Metabolic Vulnerability Are Associated with Disability and Neurodegeneration in Multiple Sclerosis"

_nutrients, 2024, doi:10.3390/nu16172866_

Round 1

Reviewer 1 Report

Comments and Suggestions for Authors

This is an interesting study correlating  blood biomarkers of inflammatory and metabolic vulnerability  and neurodegeneration in multiple sclerosis.

I have some minor suggestions, to improve :

1. Neurocognitive evaluation could be included.

2.  You mention 2010-revision of the McDonald criteria [20]. Please include the McDonald criteria-2017

3. Do you have specific data  of cerebral white matter, brainstem, cerebellum, corpus callosum, globus pallidus, thalamus volumes?

4. Discuss about other MRI techniques to demonstrate brain volumetric reduction (such as Freesurfer) and white matter integrity

Beyond traditional MRI, advanced imaging modalities provide more

information about structural white matter tract integrity and brain

function. These techniques include diffusion tensor imaging (DTI),

magnetization transfer ratio (MTR), and functional MRI (fMRI). DTI can

detect decreased microstructural integrity of both lesional and normal

appearing white matter in children with MS, based on impaired water

diffusivity. In children with MS, these microstructural alterations have

been demonstrated in NAWM, especially in the corpus callosum.

. Studies have consistently shown both decreased fractional anisotropy and more variable changes in radial and axial diffusivity, which indicate axonal loss and demyelination,respectively

Pereira FV. Pediatric inflammatory demyelinating disorders and mimickers: How to differentiate with MRI? Autoimmun Rev. 2021 May;20(5):102801.

Comments on the Quality of English Language

No specific comments.

Author Response

Comment 1.1. Neurocognitive evaluation could be included.

Response 1.1. Unfortunately, we did not have neurocognitive evaluations.

Comment 1.2.  You mention 2010-revision of the McDonald criteria [20]. Please include the McDonald criteria-2017

Response 1.2. The study was done prior to the McDonald 2017 criteria. 2010-revision of the McDonald criteria (Reference 20, now 18) is correct.

Comment 1.3. Do you have specific data of cerebral white matter, brainstem, cerebellum, corpus callosum, globus pallidus, thalamus volumes?

Response 1.3. We have data on white matter volumes, which were not presented because it is redundant given the information provided on whole brain, gray matter, and ventricular volumes. White matter volume measurements are also more difficult to interpret from a neurodegeneration point of view as it is increased by active inflammation and edema in white matter. We did not analyze data on the regional volumes of brain stem, cerebellum, corpus callosum, globus pallidus, and thalamus.

Comment 1.4. Discuss about other MRI techniques to demonstrate brain volumetric reduction (such as Freesurfer) and white matter integrity. Beyond traditional MRI, advanced imaging modalities provide more information about structural white matter tract integrity and brain function. These techniques include diffusion tensor imaging (DTI), magnetization transfer ratio (MTR), and functional MRI (fMRI). DTI can detect decreased microstructural integrity of both lesional and normal appearing white matter in children with MS, based on impaired water diffusivity. In children with MS, these microstructural alterations have been demonstrated in NAWM, especially in the corpus callosum. Studies have consistently shown both decreased fractional anisotropy and more variable changes in radial and axial diffusivity, which indicate axonal loss and demyelination, respectively. Pereira FV. Pediatric inflammatory demyelinating disorders and mimickers: How to differentiate with MRI? Autoimmun Rev. 2021 May;20(5):102801.

Response 1.4. This is added to Discussion, Page 12, Lines 345-358.

Reviewer 2 Report

Comments and Suggestions for Authors

The present study describes the association of inflammatory blood-derived blood biomarkers and metabolic vulnerability with disability and neurodegeneration in multiple sclerosis. The authors reveal the levels of multiple metabolic markers taking into account specific indexes such as Inflammatory Vulnerability Index, Metabolic Malnutrition Index and Metabolic Vulnerability Index. The manuscript is well written and the work is interesting. I have some comments and suggestions:

The authors should give characteristic NMR spectra of the related biomarkers demonstrating also differences between healthy and control subjects

Please explain in details of Expanded Disability Status Scale and the role of this approach in clinical practice.

What is the LP4 algorithm which is used for the analysis of NRM spectra?

In Figure 1 statistic should be included among control, RR and PMS.

While the strengths are evident, a more explicit discussion of the study's limitations would provide a balanced perspective (such as method selection vs. mass spectrometry, group of samples).

Some future directions could be also discussed as well as some numerical results as a conclusion might be discussed.

Author Response

Comment 2.1. The authors should give characteristic NMR spectra of the related biomarkers demonstrating also differences between healthy and control subjects.

Response 2.1. The NMR analyses were conducted at a commercial clinical laboratory. We only have quantitative measurements but do not have the individual spectra. The characteristic NMR spectra are described by Bedi et al. DOI:10.3390/jcm9092915.

Comment 2.2. Please explain in details of Expanded Disability Status Scale and the role of this approach in clinical practice.

Response 2.2. The key points in the EDSS are described in Methods, Page 3, Line 105.

Comment 2.3. What is the LP4 algorithm which is used for the analysis of NMR spectra?

Response 2.3. The expansion for LP4 is provided in Methods, Page 3, Line 135. The LP4 (or LipoProfile-4) deconvolution algorithm that is employed to obtain lipoprotein particle sizes and concentrations, glycated acute phase proteins, branched chain amino acids, citrate and other metabolites from NMR spectra.

Comment 2.4. In Figure 1 statistic should be included among control, RR and PMS.

Response 2.4. The p-values for the controls, RR and PMS are provided in Figure 1 and the other Figures.

Comment 2.5. While the strengths are evident, a more explicit discussion of the study's limitations would provide a balanced perspective (such as method selection vs. mass spectrometry, group of samples).

Response 2.5. The limitations of the MRI metrics and the selection of NMR vs. mass spectrometry are described in Discussion, Page 11.

Comment 2.6. Some future directions could be also discussed as well as some numerical results as a conclusion might be discussed.

Response 2.6. The key results and future directions are summarized in the concluding paragraph, Page 14, ¶1, Line 359.